# The Impact of the COVID-19 Pandemic on Endurance and Ultra-Endurance Running

**DOI:** 10.3390/medicina57010052

**Published:** 2021-01-09

**Authors:** Volker Scheer, David Valero, Elias Villiger, Thomas Rosemann, Beat Knechtle

**Affiliations:** 1Ultra Sports Science Foundation, 69310 Pierre-Bénite, France; volkerscheer@yahoo.com (V.S.); dalevalero@gmail.com (D.V.); 2Health Science Department, Universidad a Distancia de Madrid (UDIMA), Collado Villaba, 28400 Madrid, Spain; 3Institute of Primary Care, University of Zurich, 8091 Zurich, Switzerland; evilliger@gmail.com (E.V.); thomas.rosemann@usz.ch (T.R.); 4Medbase St. Gallen Am Vadianplatz, 9006 St. Gallen, Switzerland

**Keywords:** COVID-19, endurance running, marathon, ultra-endurance, running, sport industry

## Abstract

*Background and objectives*: The COVID-19 outbreak has become a major health and economic crisis. The World Health Organization declared it a pandemic in March 2020, and many sporting events were canceled. *Materials and Methods*: We examined the effects of the COVID-19 pandemic on endurance and ultra-endurance running (UER) and analyzed finishes and events during the COVID-19 pandemic (observation period March 2020–October 2020) to the same time period pre-COVID-19 outbreak (March 2019–October 2019). *Results:* Endurance finishes decreased during the pandemic (459,029 to 42,656 (male: 277,493 to 25,582; female 181,536 to 17,074; all *p* < 0.001). Similarly, the numbers of endurance events decreased (213 vs. 61 events; *p* < 0.001). Average marathon finishing times decreased during the pandemic in men (5:18:03 ± 0:16:34 vs. 4:43:08 ± 0:25:08 h:min:s (*p* = 0.006)) and women (5:39:32 ± 0:19:29 vs. 5:14:29 ± 0:26:36 h:min:s (*p* = 0.02)). In UER, finishes decreased significantly (580,289 to 110,055; *p* < 0.001) as did events (5839 to 1791; *p* < 0.001). Popular event locations in United States, France, UK, and Germany decreased significantly (*p* < 0.05). All distance and time-limited UER events saw significant decreases (*p* < 0.05). *Conclusions*: The COVID-19 pandemic has had a significant effect on endurance and UER, and it is unlikely that running activities return to pre-pandemic levels any time soon. Mitigation strategies and safety protocols should be established.

## 1. Introduction

The coronavirus (COVID-19) pandemic, due to the severe acute respiratory syndrome coronavirus 2 (SARS-CoV-2), has become a major health and economic crisis around the world during 2020 [1]. First reports of a new viral pneumonia appeared in December 2019 in Wuhan, China, and by January 2020, the World Health Organization (WHO) reported of the spread of a new type of Coronavirus—COVID-19 [2]. The virus rapidly expanded throughout the world, and in March 2020, the WHO declared it a pandemic [2]. Countries around the world implemented travel restrictions, closed borders, and imposed local and national lockdowns of varying degrees to reduce the spread of the virus and manage health care resources [1].

These measures also included the cancelation of mass gathering and sporting events in order to reduce and control the spread of the virus, as particularly mass gatherings and sporting events on a large scale present unique challenges to public health authorities and governments [3]. Undoubtedly, this has had adverse effects on the economy, with tourism and sport tourism being important economic sectors and likely some of the sectors most severely impacted due to lockdowns, travel restrictions, and closed borders [1,4]. The extent to which this has affected endurance and ultra-endurance running (UER) events and finishing numbers is not known.

The most prominent cancelations or (postponements) of sporting events due to the COVID-19 pandemic are the Olympic Games in Tokyo 2020 and Union of European Football Associations (UEFA) Euro 2020 Football Championships [3]. However, similarly, all other professional and amateur sports, including football, basketball, golf, athletics, triathlon etc. were initially canceled or postponed, with some professional sports (e.g., football, basketball, golf) being allowed to resume sometime later despite ongoing lockdowns or restrictions of movements under strict public health protocols [5,6].

Endurance running events in 2020 were no exception, with cancelation or postponements of big city marathons such as those in Berlin, Paris, Boston, New York, and London due to safety concerns [7]. Some events were hosted as virtual competitions, whereas others such as the Tokyo marathon 2020 were only open to elite runners [7]. This has had important economic effects on the host city, as most of these big marathons attract around 50,000 participants. For the New York City Marathon, it was estimated that, in 2014, the economic impact of the race was approximately 415 million US dollars [8]. Thus, cancelation of any of these mass gathering events will likely have an important effect on the sporting industry and the local economy. However, not only mass gathering events and prominent races were canceled due to the COVID-19 pandemic but also many smaller and local endurance events, with an impact on the local economy and the sporting community that is still not fully understood, and its effects are difficult to estimate. 

Similarly, cancelation of ultra-endurance running (UER) events occurred, such as the Ultra Trail du Mont-Blanc^®^ 2020, one of the most well-known UER event worldwide that usually attracts over 7000 participants from all around the world and with race distances of up to 170 km [9]. Other iconic races, such as the Comrades marathon in South Africa, the oldest ultramarathon, as well as the oldest 100 km race (Bieler Lauftage), the Spartathlon, a 246 km race in Greece, and many more UER were canceled or held virtually in 2020 [10,11]. The Two Oceans Marathon, a prominent race in South Africa, was also canceled in 2020, a race that attracted over 34,000 participants in 2019 [4]. Many of these events were canceled at short notice with considerable loss to the stakeholders, and in the case of the Two Oceans Marathons, the estimated loss of revenue was thought to be in the region of 2 million dollars [4]. However, considering endurance and UER events as a whole, the vast majority of races are smaller events (e.g., the Al Andalus Ultra Trail in Spain or the Isle of Wight Ultra Challenge in the UK) with fewer participant and less revenue at individual races; nevertheless, they represent an integral part of the sporting community and the wider sporting tourism branch. 

Running is an important sporting activity and one of the most popular sports worldwide that has seen an important increase in participation in endurance and UER events over the last few decades [12,13]. Therefore, it is worth a closer examination of the effects from the COVID-19 pandemic on this sporting sector.

The main aim of this study, therefore, was to explore the impact of the COVID-19 pandemic on the number of endurance and UER events and finisher numbers as well as a secondary aim to explore the age and the finishing marathon times during the first few months of the COVID-19 pandemic (March 2020–October 2020) and compare them to the same time period pre-COVID-19 pandemic (March 2019–October 2019) to evaluate the effect COVID-19 has had so far on the endurance and UER event sector. Our hypothesis was that finishes and events in endurance and UER would decrease significantly during the COVID-19 pandemic.

## 2. Materials and Methods

Marathons (distance 42.195 km) were classified as endurance running events, while UER events included running distances over marathon distance, timed-events over 6 h duration, and multi-day and multi-stage events on all running surfaces [14].

### 2.1. Ethical Procedures

This study was approved by the Institutional Review Board of Kanton St. Gallen, Switzerland, with a waiver of the requirement for informed consent of the participant as the study involved the analysis of publicly available data (EKSG 01-06-2010). The study was conducted in accordance with the recognized ethical standards according to the Declaration of Helsinki (2013).

### 2.2. Data Sampling 

Data on marathon results were obtained through the publicly available database, accessible through the website (http://www.marathonguide.com/results/). This database represents the largest marathon database in the world. Data on UER events were obtained from a publicly available database, accessible through the website of the Deutsche Ultramarathon Vereinigung (DUV) at: https://statistik.d-u-v.org/geteventlist.php. The DUV contains the largest dataset on UER worldwide and has been frequently evaluated within the scientific literature [10,15,16]. Data on race events, race location, race distance, finishing numbers, race time, sex, and nationality were, when available and accessible, analyzed for the time period since the declaration of the COVID-19 pandemic by the WHO in March 2020 [2] until the end of data sampling in October 2020 COVID-19 pandemic period (March 2020–October 2020) and compared to the same time period in the prior year, called the pre-pandemic period (March 2019–October 2019). In total, 1,192,029 finishes in 7478 events were examined. 

### 2.3. Data Analysis 

Kolmogorov–Smirnov test was applied to test for normality. Descriptive analysis was performed and presented as mean and relative (%) frequency and change. Mean marathon finishing times and age were also presented with standard deviations (SD). An independent *t*-test was used to test the differences between groups and Mann–Whitney test for not normally distributed data (pre-pandemic vs. pandemic). Statistical significance was set at 5% (*p* < 0.05). All analyses were carried out using the Python programming language (Python Software Foundation, https://www.python.org/), Google Colab notebook, and the Statistical Software for the Social Sciences (IBM SPSS v26. Chicago, IL, USA).

## 3. Results

Data and results for endurance (marathon) running are available in Table 1. The number of marathon finishes according to sexes with monthly breakdowns and percentage change during the COVID-19 pandemic was compared to the pre-pandemic period and is shown in Table 1. Monthly breakdowns were used to demonstrate the evolution of the pandemic, as at different time points, different lockdown restrictions applied throughout the world. A 10.8-fold drop could be observed in total finishes pre- to pandemic times, with almost no finishes during April/May 2020. 

The number of endurance events dropped significantly pre-pandemic to the COVID-19 pandemic period from 213 to 61 races in the database (*p* < 0.001), an approximately 3.5-fold drop. Most events were held in United States (pre-pandemic vs. pandemic 61.6% vs. 72.7%), United Kingdom (10.7% vs. 11.4%), and Canada (10.0% vs. 4.5%), and the majority of finishes originated from the United States (pre-pandemic vs. pandemic 72.0% vs. 93.5%). The ratio between finishes per event dropped from 2.155 pre-pandemic to 700 finishers/event during the COVID-19 pandemic (*p* < 0.001), suggesting that events were smaller during the pandemic, as less finishes were observed. 

The average age of finishers pre-pandemic was 47.8 ± 2.0 years and, during the pandemic, was 43.6 ± 4.3 years (*p* = 0.02), with female finishers pre-pandemic 46.0 ± 2.08 years compared to 42.6 ± 2.3 years during the pandemic (*p* = 0.01) and for male finishers 48.9 ± 2.1 years and 43.6 ± 4.3 years (*p* = 0.02), respectively. 

Average marathon finishing times for men pre-pandemic were 5:18:03 ± 0:16:34 h:min:s compared to 4:43:08 ± 0:25:08 h:min:s (*p* = 0.006) pandemic period and for women 5:39:32 ± 0:19:29 h:min:sec compared to 5:14:29 ± 0:26:36 h:min:s (*p* = 0.02), respectively. 

Data for UER finishes, event distances, and event locations are shown in Table 2, Table 3 and Table 4. Data for UER event finishes, ultra events, and finishes per event with monthly breakdowns and percentage change during the COVID pandemic compared to the pre-pandemic period are shown in Table 2. 

A 5.3-fold decrease in UER finishes can be observed during the COVID-19 pandemic, and a 3.3-fold decrease in UER events. Finishes per events also decreased from 99.8 to 53.7 finishes/event, a 1.9-fold decrease, demonstrating that events were smaller with fewer finishes. 

The 50 km distance remained the most popular UER distance, and data for UER event finishes in distance-limited events (50 km, 100 km, and 100 miles) and time-limited events (6 h, 12 h, and 24 h) with monthly breakdowns and percentage change during the COVID pandemic compared to pre-pandemic period are shown in Table 3. 

UER event locations (countries) during the COVID-19 pandemic are compared monthly to the pre-pandemic period and shown in Table 4. The three top event locations pre-pandemic were USA, France, and United Kingdom and during the COVID-19 pandemic were USA, Germany, and United Kingdom. For further comparison and illustration, the UER events of China Tapei are shown, as this location showed, in contrast to others, a relative increase during the COVID-19 pandemic, especially in April 2020 with over 90% of events hosted in this location. 

Additionally, UER event numbers and performances from 2018 (6708 vs. 609,847) increased to 2019 (7468 vs. 671,738), increases of 11.3% and 10.1%, respectively.

## 4. Discussion

Running is one of the most popular sports worldwide, with endurance and UER showing important increases in participation and finishes over the last few decades [13,14]. Since the onset of the COVID-19 pandemic in March 2020 [2], many sporting events were canceled or postponed, but the impact of the pandemic on endurance and UER has not been examined thus far.

The aim of the study was, therefore, to explore the impact of the COVID-19 pandemic on endurance and UER events and participation and the implications of the endurance and UER sector as a whole. We hypothesized that, during the COVID-19 pandemic, finishing numbers and events would decrease significantly when compared to the same time period in the preceding year in pre-pandemic times.

The main findings of our study were: (i) finishes in endurance races decreased significantly during the pandemic, with an almost 11-fold decrease; (ii) event numbers decreased significantly in endurance events during the pandemic, with almost no activity during April/May 2020; finishes/event ratio decreased as well, suggesting that events were smaller during the pandemic, as less finishes were observed; (iii) average age of endurance finishers decreased significantly during the pandemic, as did the marathon finishing times; (iv) finishes in UER decreased significantly during the pandemic, with an over 5-fold decrease, with the biggest decrease in April/May 2020; (v) event numbers and locations in UER decreased significantly, with a proportional increase in events in China Tapei during the pandemic, especially in April 2020; (vi) the 50 km event remained the most popular distance based UER event and the 6 h the most popular time-limited event, both showing a significant decline during the pandemic.

As with other sports and sporting events during the global crisis, many endurances and UER were postponed, canceled, or held virtually due the continued uncertainty of the virus’s spread and the potential risk of spreading the virus through congregation of runners. From personal experience, a number of events were held virtually, however, we are unaware of any available data sets on a larger scale for analysis and comparison.

Our data show the effect of the COVID-19 pandemic since it was first declared on 11 March 2020 [2] on endurance and UER running. We observed a significant decrease in finishes and event numbers in endurance and UER, especially in the first two full months after the pandemic was declared (April/May 2020) with little to no activity in endurance and UER during this time. This is something that could be expected, as with national and local lockdowns, restrictions, and bans on travel, very little movement occurred during these time periods [1,3]. Overall finishes in endurance running decreased almost 11-fold, while UER finishes decreased 5-fold during the observation period of the pandemic. One explanation may be that endurance running is generally more popular than UER and that the UER community tends to be smaller. Perhaps participation continued in smaller, more local races, as demonstrated by our data that showed that approximately 50 finishes/event in UER were observed compared to 700 finishes/endurance event, although the percentage drop in events for endurance and UER were very similar. Additionally, the number of events in the endurance running database was comparatively smaller than for UER events, which further may add to this. UER events and finishing numbers have been increasing over the last 20 years [12,13], and this can similarly be observed when comparing UER event numbers and performances from 2018 to 2019. A further increase in 2020 could have been expected if not for the COVID-19 pandemic.

Another interesting observation in endurance events is that the average age of marathon finishers decreased during the pandemic, as did the average marathon finishing times. The reason for this may be, that more experienced runners kept participating in marathons during the COVID-19 pandemic, whilst more amateur runners stayed at home, similarly to older participants, that present a higher risk population of developing more severe symptoms of COVID-19.

In UER, race distances of 50 km are generally the most popular UER distances [13]; this was also observed during the pandemic, however, with a significant decrease compared to pre-pandemic levels. The same could be observed in all other distance limited events (100 km and 100 miles) and time limited UER (6 h, 12 h, and 24 h). Since August 2020, numbers of finishes have increased notably, however are still lagging behind considerably compared to pre-pandemic levels. 

Running events can have positive economic effects with short and long term economic consequences [17], whereas cancelation may have a detrimental effect, as exemplified by the cancelation of the Two Oceans Marathon and the New York Marathon [4,8]. The cancelation of the 2020 Two Oceans Marathon in South Africa reported an approximate loss of revenue of around 2 million dollars [4]. Similarly, the New York City marathon incurred significant losses when canceled due to the effects of a devastating natural disaster (hurricane Sandy) with estimated losses of charitable donations of 36.1 million US dollars and an total estimated economic impact of the race of approximately 415 million US dollars in 2014 [8]. Additionally important to note is that running events can have positive economic effects during the COVID-19 pandemic, as sporting events can create positive publicity in sending out the signal that the city or the country is open for business and thus can create economic growth [17]. This was the case of China Tapei that was the most popular UER event location in April/May 2020. Whether this also created additional economic growth is not known. 

However, it is also important to note that running poses an extremely low risk of COVID-19 transmission, with only one reported case among 571,401 athletes and 98,035 officials and staff that took part in 787 races and track meetings in Japan since July 2020 [18]. All events were held without spectators, and specific safety measures were introduced, which may potentially allow running activities to resume with appropriate safety protocols [18]. Such tools have been developed by the WHO that provides a risk assessment tool for sporting and mass gathering events during the COVID-19 pandemic, considering specific action plans and risk mitigation strategies [19]. These strategies may help in delaying the spread of an outbreak [20] and may be useful tools in the decision-making processes of hosting an event [3]. 

Our results show the devastating effects COVID-19 had on endurance and UER. It is necessary to examine the possibility of returning to pre-COVID-19 levels, as a whole branch of the sporting industry is dependent on this activity. With risk assessment tools, mitigation strategies, and strict safety protocols, a gradual return to endurance and UER may be possible, especially considering the extremely low risk outdoor running poses for contracting COVID-19. However, a return to pre-pandemic levels any time soon remains unlikely until the time an effective drug treatment or vaccine becomes available [20]. 

Further studies examining the economic effect of the COVID-19 pandemic on endurance and UER may be useful to estimate the potential loss to the industry in addition to examining the impact on health. Similarly, examining the demographics and the performance times further and over a longer time period may provide additional important information on how COVID-19 has impacted running and UER.

### Limitations

Of the two publicly available databases used for this analysis, the marathon results database (http://www.marathonguide.com/results/) is the largest database of marathon results in the world, fully searchable by name, place, or time. However, the majority of data pertain to races held in the United States, Canada, Australia, and New Zealand, and we recognize this as a limiting factor for applicability on races worldwide. The DUV database (https://statistik.d-u-v.org/geteventlist.php) is the largest database worldwide of UER events and has been widely used in the past in the scientific literature [10,15,21]. However, as with any large database, not all results may be complete, and we recognize that this as a limiting factor. Nevertheless, the aim of the study was to examine the impact of COVID-19 on endurance and UER, and both databases provide a sufficiently sizable data sample for analyses. Examining and comparing marathon finishing times pre-pandemic to pandemic provides some interesting insights, however, comparing several different events with different ambient conditions and race profiles has its limitations and needs to be interpreted with care.

## 5. Conclusions

Endurance and UER have seen a significant decrease in the number of finishes and events during the COVID-19 pandemic with a devastating effect on the sporting industry. It is unlikely that running activities will return to pre-pandemic levels any time soon, and mitigation strategies and safety protocols should be established until the time an effective drug treatment or vaccine becomes available. Future studies might analyze the economic impact COVID-19 has had on the endurance and UER industry as a whole.

## Figures and Tables

**Table 1 medicina-57-00052-t001:** Data on number of marathon finishes according to sexes with monthly breakdowns and percentage change during the time period of the start of the COVID pandemic (March 2020) until the end of the observation period (October 2020) and comparison to the same time period pre-COVID in 2019 (March–October 2019).

	March	April	May	June	July	August	September	October	Total
**Marathon finishes 2019**	45,593	100,898	66,159	28,176	15,633	11,512	26,996	164,062	459,029
**Marathon finishes 2020**	32,549	8	0	199	293	1872	4676	3059	42,656 *
Change (%)	−28.6	−100.0	−100.0	−99.3	−98.1	−83.7	−82.7	−98.1	−90.7
**Male finishes 2019**	27,812	58,521	41,680	17,187	10,161	8058	17,277	96,797	277,493
**Male finishes 2020**	19,204	4	0	137	170	1179	2899	1988	25,582 *
Change (%)	−31.0	−100.0	−100.0	−99.2	−98.3	−85.4	−83.2	−97.9	−93.0
**Female finishes 2019**	17,781	42,377	24,479	10,989	5472	3454	9719	67,265	181,536
**Female finishes 2020**	13,345	4	0	62	123	693	1777	1071	17,074 *
Change (%)	−24.9	−100.0	−100.0	−99.4	−97.8	−79.9	−81.7	−98.4	−92.8

* *p* < 0.001.

**Table 2 medicina-57-00052-t002:** Data for ultra-endurance event finishes, ultra events, and finishes per event with monthly breakdowns and percentage change during the time period of the start of the COVID pandemic (March 2020) until the end of the observation period (October 2020) and comparison to the same time period pre-COVID in 2019 (March–October 2019).

	March	April	May	June	July	August	September	October	Total
**Finishes Ultra 2019**	56,741	96,709	74,678	107,273	58,196	54,627	62,147	69,927	580,289
**Finishes Ultra 2020**	21,310	680	1262	3031	10,124	20,978	27,860	24,810	110,055 *
Change (%)	−62.4	−99.3	−98.3	−97.2	−82.6	−61.6	−55.2	−64.5	−81.0
**Ultra events 2019**	577	643	775	883	648	707	799	807	5839
**Ultra events 2020**	205	21	49	92	175	360	447	442	1791 *
Change (%)	−64.5	−96.7	−93.7	−89.6	−73.0	−49.1	−44.1	−45.2	−69.3
**Finishes/event 2019**	98.3	150.4	96.4	121.5	89.8	77.3	77.8	86.7	99.8
**Finishes/event 2020**	104.0	32.4	25.8	32.9	57.9	58.3	62.3	56.1	53.7 ^#^
Change (%)	5.7	−78.5	−73.3	−72.9	−35.6	−24.6	−19.9	−35.2	−46.2

* *p* < 0.001, ^#^
*p* < 0.05.

**Table 3 medicina-57-00052-t003:** Data for ultra-endurance event finishes in distance-limited events (50 km, 100 km, and 100 miles) and time-limited events (6 h, 12 h, and 24 h) with monthly breakdowns and percentage change during the time period of the start of the COVID pandemic (March 2020) until the end of the observation period (October 2020) and comparison to the same time period pre-COVID in 2019 (March–October 2019).

Finishes	March	April	May	June	July	August	September	October	Total
**50 km 2019**	19,289	36,907	17,831	14,433	10,126	10,206	6554	15,424	130,770
**50 km 2020**	5626	32	126	390	974	3588	4921	5189	20,846 ^†^
Change (%)	−70.8	−99.9	−99.3	−97.3	−90.4	−64.8	−24.9	−66.4	−84.1
**100 km 2019**	6690	5013	2848	8915	5097	3112	7819	9059	48,553
**100 km 2020**	250	19	205	149	467	992	3782	1240	7104 ^#^
Change (%)	−96.3	−99.6	−92.8	−98.3	−90.8	−68.1	−51.6	−86.3	−85.4
**100 miles 2019**	1364	604	1521	3113	1103	3092	2484	3257	16,538
**100 miles 2020**	129	0	41	34	604	289	1340	1048	3485 ^†^
Change (%)	−90.5	−100.0	−97.3	−98.9	−45.2	−90.7	−46.1	−67.8	−78.9
**6 h 2019**	4902	4175	3621	2206	2587	1617	8292	3043	30,443
**6 h 2020**	2374	0	0	282	460	717	936	2917	7686 ^†^
Change (%)	−51.6	−100.0	−100.0	−87.2	−82.2	−55.7	−88.7	−4.1	−74.8
**12 h 2019**	1329	3825	3477	3673	3406	1586	484	3919	21,699
**12 h 2020**	143	0	84	246	761	646	1334	1717	4931 ^†^
Change (%)	−89.2	−100.0	−97.6	−93.3	−77.7	−59.3	175.6	−56.2	−77.3
**24 h 2019**	1753	905	6617	4940	466	3645	865	1910	21,101
**24 h 2020**	243	0	162	24	1677	1043	1540	895	5584 ^†^
Change (%)	−86.1	−100.0	−97.6	−99.5	259.9	−71.4	78.0	−53.1	−73.5

^#^*p* < 0.001, ^†^
*p* < 0.05.

**Table 4 medicina-57-00052-t004:** Event location (country) of ultra-endurance running events pre- COVID-19 (March 2019-Ocotber 2019) compared with monthly numbers and total numbers and percentage change to the observation period during the COVID-19 pandemic (March 2020–October 2020) with listings of the top three event locations (pre-COVD-19: USA, FRA (France), and GBR (Great Britain); COVID-19 pandemic: USA, GER (Germany), and UK (United Kingdom)), with percentage of all event locations during that particular month). Further analysis of the location with the greatest change during the observation period is included (TPE: China Tapei).

Ultra Event Location	March	April	May	June	July	August	September	October	Total
**2019 USA (%)**	183 (31.7)	214 (33.3)	210 (27.1)	199 (22.5)	161 (24.8)	216 (30.6)	217 (27.2)	266 (33.0)	1666 (28.5)
**2020 USA (%)**	84 (41.0)	0 (0.0)	13 (26.5)	29 (31.5)	58 (33.1)	80 (22.2)	126 (28.2)	173 (39.1)	563 (31.4) *
Change (%)	−54.1	−100.0	−93.8	−85.4	−64.0	−63.0	−41.9	−35.0	−66.2
**2019 FRA (%)**	30 (5.2)	67 (10.4)	65 (8.4)	103 (11.7)	68 (10.5)	35 (5.0)	43 (5.4)	58 (7.2)	469 (8.0)
**2020 FRA (%)**	15 (7.3)	0 (0.0)	0 (0.0)	0 (0.0)	4 (2.3)	33 (9.2)	25 (5.6)	21 (4.8)	98 (5.5) ^#^
Change (%)	−99.8	N/A	N/A	N/A	−99.6	−99.9	−99.9	−99.9	−100.0
**2019 UK (%)**	44 (7.6)	28 (4.4)	64 (8.3)	63 (7.1)	50 (7.7)	54 (7.6)	86 (10.8)	32 (4.0)	421 (7.2)
**2020 UK (%)**	11 (5.4)	0 (0.0)	0 (0.0)	1 (1.1)	3 (1.7)	17 (4.7)	31 (7.0)	36 (8.1)	99 (5.5) *
Change (%)	−99.8	N/A	N/A	−98.4	−99.3	−99.9	−100.0	−99.9	−100.0
**2019 GER (%)**	30 (5.2)	23 (3.6)	30 (3.9)	50 (5.7)	27 (4.3)	58 (8.2)	37 (4.6)	23 (2.9)	278 (4.8)
**2020 GER (%)**	5 (2.4)	0 (0.0)	4 (8.2)	11 (12.0)	13 (7.4)	30 (8.3)	29 (6.5)	27 (6.1)	119 (6.6) ^#^
Change (%)	−83.3	−100.0	−86.7	−78.0	−51.9	−48.3	−21.6	17.4	−57.2
**2019 TPE (%)**	20 (3.5)	24 (3.7)	13 (1.7)	20 (2.3)	12 (1.9)	18 (2.5)	15 (1.9)	16 (2.0)	138 (2.4)
**2020 TPE (%)**	10 (4.9)	19 (90.5)	17 (34.7)	12 (13.0)	12 (6.9)	19 (5.3)	14 (3.1)	6 (1.4)	109 (6.1) ^#^
Change (%)	−50.0	−20.8	30.8	−40.0	0.0	5.6	−6.7	−62.5	−21.0

* *p* < 0.001, ^#^
*p* < 0.05.

## Data Availability

Not applicable.

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
