# Peer review of "The Impact of the COVID-19 Pandemic on Endurance and Ultra-Endurance Running"

_medicina, 2021, doi:10.3390/medicina57010052_

Round 1

Reviewer 1 Report

Overall this is a well-executed study, but I am unsure of the novelty or scientific merit. I think most would agree that participation in nearly every type of recreational activity has been reduced during COVID. In this absence of anchoring these findings to real economic fall out or potentially poorer health outcomes, I do not believe this work represents a significant contribution to the field. I have a few specific comments below but overall would recommend the authors revise to better address the "so what" aspect of this work, that is to say that organized running events saw up to 100% reduction in participation during COVID...so what? I am not sure that knowing running participation was decreased is helpful by itself. If the authors can adequately address that in future drafts, I think this work could be suitable for publication. 

Abstract: No comments, clearly summarizes the manuscript

Introduction: 

  • Could be more focused. I was unclear on what direction this manuscript was going until the last 2 paragraphs of the introduction. Indicating if the purpose of this manuscript was to document the economic or health impact on individuals of reduced endurance events (or both) would be helpful.
  • The current purpose and hypothesis seem very obvious - everything was reduced during COVID - and not sufficient on their own as justification for a manuscript. 

Methods:

  • Methods seem appropriate for the study. Chi-square analysis between sexes and years may add an additional strength to analysis. 
  • Small correction to statistical analysis sentence - SPSS is the Statistical Package for the Social Sciences by IBM, not IMB. 

Results: 

  • Results are generally clear and well suited for presentation in tables. 
  • Tables: The p-value thresholds seem a bit arbitrary - for example the difference between 0.001 and 0.002 is unlikely to be clinically/practically meaningful. While they may all be correct, I would encourage the authors to use more conventional thresholds that may indicate differences more generally (such as p < 0.001, 0.01, 0.05) and confirm that the keys for all tables are correct since the p-values are not in increasing order. For example, in table 2 I would expect 0.004 to be presented between 0.001 and 0.04 and table 4 I would expect 0.003 to follow 0.002. 

Discussion: 

  • Much of the discussion is somewhat of a reiteration of the introduction. 
  • I wonder if the authors know or can comment on the possibility of post-COVID results in the databases being for "virtual races", i.e. people run the same distance as the real race but can choose their own route and enter their results manually.

Author Response

Dear reviewer,

We thank you for your very valuable feedback and suggestions. Please find below our point-by-point answers. We hope this clarifies your queries and we belief our manuscript has been strengthened by your comments. Thank you.

Overall this is a well-executed study, but I am unsure of the novelty or scientific merit. I think most would agree that participation in nearly every type of recreational activity has been reduced during COVID. In this absence of anchoring these findings to real economic fall out or potentially poorer health outcomes, I do not believe this work represents a significant contribution to the field. I have a few specific comments below but overall would recommend the authors revise to better address the "so what" aspect of this work, that is to say that organized running events saw up to 100% reduction in participation during COVID...so what? I am not sure that knowing running participation was decreased is helpful by itself. If the authors can adequately address that in future drafts, I think this work could be suitable for publication. 

Reply: Thank you. We respectfully disagree with the comment about novelty and scientific merit. Although the reduction in events during the COVID seems obvious, there is no scientific data to back this and corroborate this. We believe these data are therefore important, as they provide a scientific basis for discussions, instead of anecdotal evidence or personal impressions.

We have provided examples of the economic impact where scientific data are available but are unable to provide or speculate of the complete fallout to the whole industry, as this is out with the scope of this article. This however is a very interesting point and would be an interesting follow up study, based on the scientific data presented in this article. Similarly, we have no data or can speculate on potentially poorer health outcomes because of cancellation of races, again an interesting point but out with the scope of the study. It is possible that athletes continued to train or participate in virtual races, however we have no data to corroborate this. But maybe this is another study for future directions (questionnaire based?)

But we thank you for your suggestions and have added this to the manuscript for future outlooks:

‘Further studies examining the economic effect of the COVID-19 pandemic on endurance and UER may be useful to estimate the potential loss to the industry, as well as examining the impact on health.’

Abstract: No comments, clearly summarizes the manuscript

Reply: Thank you.

Introduction: 

  • Could be more focused. I was unclear on what direction this manuscript was going until the last 2 paragraphs of the introduction. Indicating if the purpose of this manuscript was to document the economic or health impact on individuals of reduced endurance events (or both) would be helpful.

Reply: Thank you. We have introduced the general aim of the study earlier in the introduction and also clarified the aims.

Further please see detailed comment above. Examples of the impact of the economy have been provided where scientific data were available. It is out with the scope of this article to calculate or speculate of the extent of economic fallout for the whole industry, but we agree this may be an interesting study to conduct on the basis of the scientific data of the present study. Similarly, the impact on health is an interesting point and is worthy of further assessment, but out with the aims and scope of this work. Again, current data may provide the scientific background for this problem, however it is unclear of how many of the individuals participated in virtual races or continued their normal exercise routine, therefore the impact on health may be minimal or difficult to calculate.

We have added your suggestions in the outlook for further studies in this area.

‘Further studies examining the economic effect of the COVID-19 pandemic on endurance and UER may be useful to estimate the potential loss to the industry, as well as examining the impact on health.’

  • The current purpose and hypothesis seem very obvious - everything was reduced during COVID - and not sufficient on their own as justification for a manuscript. 

Reply: Thank you. Please see detailed comment above. We believe the presented data is novel and provides a scientific basis of the issue at hand and as outlined previously and may be helpful for further analysis on economy or health. Although personal impression of the amount of reduction of events and numbers may be widespread, this provides no reliable or scientific data, which we provide with a worldwide analysis. 

Methods:

  • Methods seem appropriate for the study. Chi-square analysis between sexes and years may add an additional strength to analysis. 

Reply:  Thank you. We have clarified the statistical analyses section as per reviewer 2 and the methods used.

  • Small correction to statistical analysis sentence - SPSS is the Statistical Package for the Social Sciences by IBM, not IMB. 

Reply: Thank you- well spotted spelling mistake- changed.

Results: 

  • Results are generally clear and well suited for presentation in tables. 

Reply: Thank you.

  • Tables: The p-value thresholds seem a bit arbitrary - for example the difference between 0.001 and 0.002 is unlikely to be clinically/practically meaningful. While they may all be correct, I would encourage the authors to use more conventional thresholds that may indicate differences more generally (such as p < 0.001, 0.01, 0.05) and confirm that the keys for all tables are correct since the p-values are not in increasing order. For example, in table 2 I would expect 0.004 to be presented between 0.001 and 0.04 and table 4 I would expect 0.003 to follow 0.002. 

Reply: Thank you. This has been changed as suggested.

Discussion: 

  • Much of the discussion is somewhat of a reiteration of the introduction. 
  • I wonder if the authors know or can comment on the possibility of post-COVID results in the databases being for "virtual races", i.e. people run the same distance as the real race but can choose their own route and enter their results manually.

Reply: Thank you. Unfortunately, we are not aware of any larger database that would provide information on virtual races for analysis. We are in agreement, that from personal experience a number of events were held virtually, and again from personal experience participation numbers in some of these events was much lower than during normal times. We have added this to the discussion:

 ‘From personal experience a number of events were held virtually, however we are unaware of any available data sets on a larger scale for analysis and comparison.’

We have further added to discussion your suggestion for future studies:

 ‘Further studies examining the economic effect of the COVID-19 pandemic on endurance and UER may be useful to estimate the potential loss to the industry, as well as examining the impact on health.’

Reviewer 2 Report

There are no major issues with writing mechanics or content with this manuscript. The authors have prepared an informative product.

This manuscript is unique in several ways. One recurring issue though, is this is really a descriptive study versus an original research style project. We already know what the answer is to the question. Anecdotal observations by any runner that regularly signs up for marathon or longer distance running events will confirm that many annual events are not occurring during the pandemic. I have been a race director the last 8 years and unfortunately, have been forced to cancel my own event.

I don’t think this means the manuscript is unworthy of consideration for publication, but that maybe some alteration in direction could improve the value of the work.

Abstract

There was only 1 month that events really increased for Taiwan. It was early, and I would guess this was due to events that were postponed at the beginning of COVID being rescheduled. It seems like this space could be better suited to other outcomes (e.g. finishing times).

Intro

No major concerns.

Methods/Results

I am most confused by the data analysis section. Based on the title and abstract, number of events and number of finishers in said events is your primary outcome. Whether compared by a single month, for the entire testing period, or between sexes, there should be no SD for these data. Tables 1-4 all report p values, but I do not see a description of a test that would have been used to analyze these non-parametric data? Could you please give more details on this?

Data is not typically replicated in a manuscript. Is there a reason for duplication of data in Table 1 and Figure 1? Finished by month are already presented in Table 1. Why repeat?

As stated above, I don’t think the interesting questions is did the number of events/finishers change? Could you get rid of the data by month altogether? This would make a much less cumbersome presentation of data without sacrificing the primary study purpose in my opinion.

To me the most interesting data is presented in Lines 136-142. I think demographics of finishers and race performance is much more interesting than whether races/finishers decreased. You have a more firm grasp than me on what publicly available data could be used for exploring these twos areas. I would highly suggest expansion of this data and including it in table or figure format. Would it be possible to present 2018 data also so a year-to-year references without COVID-19 could be made? To reduce cofactor influence, what about presenting demographic and performance data for only marathons that were held all 3 years? Do you think more elite runners continuing to participate dropped race times or maybe just a sharp drop in the most elderly/slowest runners? Maybe something else?

Discussion

Some great points are made. I would really like to see if methods/results are altered before providing in-depth comments.

I hope these comments are helpful. I look forward to seeing your revised manuscript.

Author Response

Dear reviewer,

We thank you for your very valuable feedback and suggestions. Please find below our point-by-point answers. We hope this clarifies your queries and we belief our manuscript has been strengthened by your comments. Thank you.

There are no major issues with writing mechanics or content with this manuscript. The authors have prepared an informative product.

This manuscript is unique in several ways. One recurring issue though, is this is really a descriptive study versus an original research style project. We already know what the answer is to the question. Anecdotal observations by any runner that regularly signs up for marathon or longer distance running events will confirm that many annual events are not occurring during the pandemic. I have been a race director the last 8 years and unfortunately, have been forced to cancel my own event.

I don’t think this means the manuscript is unworthy of consideration for publication, but that maybe some alteration in direction could improve the value of the work.

Reply: Thank you. We completely agree. Being race directors ourselves and involved in many events the COVID-19 pandemic has had serious effects on our industry. Although intuitive and personal experience and anecdotal observations are clear, that events were canceled during the pandemic, this study provides the scientific data on this on a worldwide scale. Similarly, our data can provide useful information for follow up studies to examine the economic or health impact.

Abstract

There was only 1 month that events really increased for Taiwan. It was early, and I would guess this was due to events that were postponed at the beginning of COVID being rescheduled. It seems like this space could be better suited to other outcomes (e.g. finishing times).

Reply: Thank you. We have deleted this and added finishing times as suggested.

Intro

No major concerns.

Reply: Thank you.

Methods/Results

I am most confused by the data analysis section. Based on the title and abstract, number of events and number of finishers in said events is your primary outcome. Whether compared by a single month, for the entire testing period, or between sexes, there should be no SD for these data. Tables 1-4 all report p values, but I do not see a description of a test that would have been used to analyze these non-parametric data? Could you please give more details on this?

Reply: Thank you- we have clarified the statistical section as suggested:

Kolmogorov-Smirnov test was applied to test for normality. Descriptive analysis was performed and presented as mean and relative (%) frequency and change. Mean marathon finishing times and age were also presented with standard deviations (SD). An independent t-test was used to test the differences between groups and Mann Whitney test for not normally distributed data (pre-pandemic vs pandemic). Statistical significance was set at 5% (P < 0.05). All analyses were carried out using the Python programming language (Python Software Foundation, https://www.python.org/), Google Colab notebook and the Statistical Software for the Social Sciences (IBM SPSS v26. Chicago, Ill, USA).’

Data is not typically replicated in a manuscript. Is there a reason for duplication of data in Table 1 and Figure 1? Finished by month are already presented in Table 1. Why repeat?

Reply: Thank you- We have deleted Figure 1.

As stated above, I don’t think the interesting questions is did the number of events/finishers change? Could you get rid of the data by month altogether? This would make a much less cumbersome presentation of data without sacrificing the primary study purpose in my opinion.

Reply: Thank you: The rationale for presenting the data by months is, that it shows, more clearly how the pandemic developed, and the effect it had on event numbers/ finishers by months and during the observation period. For example, lockdown measures and the spread of the virus were different at different time points in different countries, as outlined exemplary in the case of China Tapai. We have therefore opted to remain the data presentation in its current form as it provides additional important information.

To clarify this, we have added the following to the manuscript:

‘Monthly breakdowns were used to demonstrate the evolution of the pandemic, as at different time points, different lockdown restrictions applied throughout the world.’

To me the most interesting data is presented in Lines 136-142. I think demographics of finishers and race performance is much more interesting than whether races/finishers decreased. You have a more firm grasp than me on what publicly available data could be used for exploring these twos areas. I would highly suggest expansion of this data and including it in table or figure format. Would it be possible to present 2018 data also so a year-to-year references without COVID-19 could be made? To reduce cofactor influence, what about presenting demographic and performance data for only marathons that were held all 3 years? Do you think more elite runners continuing to participate dropped race times or maybe just a sharp drop in the most elderly/slowest runners? Maybe something else?

Reply: Thank you. This is an interesting aspect. Data on demographics is accessible through the publicly available website for UER at DUV (http://statistik.d-u-v.org/index.php) for each single race (e.g., JFK 50 mile race- held in Nov 2020 (http://statistik.d-u-v.org/getresultevent.php?event=62895), which could be analysed and compared to the same specific race on prior years, however data for analysis of all events unfortunately is not accessible nor feasible, and out with the scope of our study.

I think all your points and suggestions for further analyses specifically for marathons are very interesting and could be something for a future study but within the current aims unfortunately this is out with the scope of the manuscript.

We have added your suggestion to the results and discussions as well as the information of events/ finishers, that were publicly available for 2018 for comparison:

‘Additionally, UER event numbers and performances from 2018 (6708 vs 609847) increased to 2019 (7468 vs 671738), an increase of 11.3% and 10.1%, respectively.’

‘Similarly, examining the demographics and performance times further and over a longer time period, may provide additional important information on how COVID-19 has impacted on running and UER’.

‘UER events and finishing numbers have been increasing over the last 20 years [12,13] and similarly this can be observed when comparing UER event numbers and performances from 2018 to 2019. A further increase in 2020 could have been expected, if not for the COVID-19 pandemic.’

An additional point about examining marathon performances: There are several studies that have examined, marathon performances, in specific races or age groups (some from our study group), e.g:

Knechtle B, Di Gangi S, Rüst CA, Villiger E, Rosemann T, Nikolaidis PT. The role of weather conditions on running performance in the Boston Marathon from 1972 to 2018. PLoS One. 2019 Mar 8;14(3):e0212797. doi: 10.1371/journal.pone.0212797. eCollection 2019.

Nikolaidis PT, Di Gangi S, Chtourou H, Rüst CA, Rosemann T, Knechtle B. The Role of Environmental Conditions on Marathon Running Performance in Men Competing in Boston Marathon from 1897 to 2018. Int J Environ Res Public Health. 2019 Feb 20;16(4):614. doi: 10.3390/ijerph16040614.

And performances vary according to race, race profile, ambient conditions etc., so examining global times and comparing them to different years may be difficult. Again, we believe this may be a worthy study to conduct on performances but has several limitations as outlined above.

As we recognize this, we have added to the limitation section, that examining finishing times is an interesting aspect but has several limitations:

‘Examining and comparing marathon finishing times pre-pandemic to pandemic provides some interesting insights, however by comparing several different events, with different ambient conditions and race profiles has its limitations and needs to be interpreted with care.’

Discussion

Some great points are made. I would really like to see if methods/results are altered before providing in-depth comments.

I hope these comments are helpful. I look forward to seeing your revised manuscript.

Reply: Thank you for your insightful review and suggestions. We hope we have answered all your queries and thank you for the great suggestions for future studies/ analyses.

Round 2

Reviewer 1 Report

Thank you for the opportunity to review this revision. I believe the authors have adequately addressed my concerns and strengthened their manuscript while not overstating their findings. I do not have any additional comments.